# Smoking Cessation during the Second Half of Pregnancy Prevents Low Birth Weight among Australian Born Babies in Regional New South Wales

**DOI:** 10.3390/ijerph18073417

**Published:** 2021-03-25

**Authors:** Pramesh Raj Ghimire, Julie Mooney, Louise Fox, Lorraine Dubois

**Affiliations:** 1Population Health, Southern New South Wales Local Health District, Queanbeyan, NSW 2620, Australia; Lorraine.Dubois@health.nsw.gov.au; 2Nursing and Midwifery, Southern New South Wales Local Health District, Queanbeyan, NSW 2620, Australia; Julie.Mooney@health.nsw.gov.au; 3Integrated Care and Allied Health, Southern New South Wales Local Health District, Queanbeyan, NSW 2620, Australia; Lou.Fox@health.nsw.gov.au

**Keywords:** tobacco smoking, epidemiology, smoking cessation, low birth weight, pregnancy, Australia

## Abstract

Smoking during pregnancy is a modifiable risk behavior of adverse health outcomes including low birth weight (LBW), and LBW is a key marker of newborns immediate and future health. This study aimed to examine the association between smoking cessation during the second half of pregnancy and LBW among babies born in Southern New South Wales Local Health District (SNSWLHD). Routinely collected perinatal data on singleton live births for the period 2011–2019 in five public hospitals of SNSWLHD were utilized. Multivariate logistic regression models were fitted to examine the association between smoking cessation during the second half of pregnancy and LBW. Analyses showed that mothers who ceased smoking during the second half of pregnancy were 44% less likely to have LBW babies (adjusted odds ratio (aOR) = 0.56; 95% confidence interval (CI): 0.34, 0.94) compared to those who continued smoking throughout pregnancy. Mothers who reported an average daily dose of 1–10 or >10 cigarettes during the second half of pregnancy were significantly more likely to have babies with LBW than those who ceased smoking during the second half of pregnancy. Early identification of smoking behavior and promotion of smoking-cessation intervention for risk populations including pregnant women within the older age bracket (35–49 years) is imperative to reduce LBW.

## 1. Introduction

Babies with <2500 g of birth weight (regardless of the gestational age) are considered to have low birth weight (LBW) [1,2]. LBW constitutes wide-ranging adverse health and economic consequences. Future chronic health conditions such as diabetes [3], cardiovascular diseases [4], asthma [5,6] and renal diseases [7] are more prevalent among LBW babies. Adults born with a LBW are predisposed to poor neuromotor functioning [8], unemployment, and relying on social benefits longer term [9]. The admission rate to neonatal intensive care units (NICUs) for LBW babies is higher than those of normal birth weight [10], putting an extra burden on health care systems.

The association between tobacco smoking in pregnancy and LBW has been well documented in public health literature. For example, a meta-analysis of studies conducted in the Americas [11] concluded that mothers who are active smokers during pregnancy have double the likelihood of delivering LBW babies. An Australian study that used New South Wales (NSW) Midwives Data Collection also confirmed that babies born to mothers who smoked during pregnancy were at higher risk of having LBW [12]. Research conducted in developed countries including Australia also suggested that smoking during pregnancy is a dose-dependent risk factor in that the risk of LBW is elevated with increased number of cigarettes [13,14,15].

Even though the relationship between smoking in pregnancy and LBW has been clearly established in scientific literature, the timing of smoking cessation in pregnancy that can prevent LBW appears to be complex. This complexity often hinders the design and implementation of smoking-cessation strategies in the health care services where the guideline and time frame for pregnancy care is always bound by a short window of opportunity. An American study found that the risk of a LBW infant for mothers who quit smoking in the first trimester was lower compared to that of mothers who reported smoking beyond the first trimester [16]. Batech and colleagues found that mothers who quit smoking throughout pregnancy were less likely to have LBW babies compared to mothers who continued smoking during pregnancy [17]. A quasi-experimental study of Swedish Medical Birth Register data highlighted the positive association between improved birth weight and smoking cessation in the third trimester [18]. An Australian study by Chan et al. [13] that used data of teenage mothers found that smoking cessation in the second half of pregnancy reduced LBW.

To reduce smoking, NSW Smoking Cessation Framework has long emphasized the 5As model of brief intervention (asking, advising, assessing, assisting, and arranging for follow up) in health services [19]; the NSW tobacco strategy (2012–2021) [20] highlights the responsibility of NSW Health to develop strategies to promote smoking-cessation intervention for pregnant women. The NSW Maternal & Child Health Care Policy [21] also emphasized that the optimal level of maternal and newborn health can be achieved by preventing risk and promoting protective factors while implementing health promotion initiatives.

Notwithstanding the established smoking-cessation framework and policy guidelines around improving newborn health, the latest health data published by NSW government reported that smoking during pregnancy in recent years has remained stable ranging from 9.1% in 2014 to 9.3% in 2018 [22]. Similarly, the NSW Mothers and Babies report published in 2018 [23] indicated that LBW in New South Wales in the same period has also remained stable (6.3% in 2014 to 6.8% in 2018). This study aimed to estimate trends and examine the impact of smoking cessation during the second half of pregnancy on LBW among mothers of reproductive age (15–49 years). The findings from this study can be beneficial for policy makers, health managers, and clinicians to understand the risk and the protective factors for LBW and incorporate relevant strategies into policy and practice while providing pregnancy care in health services across rural and regional Local Health Districts (LHDs).

## 2. Materials and Methods

### 2.1. Data Source and Sample

The data utilized in this study were derived from New South Wales Perinatal Data Collection (PDC). PDC is a population-based surveillance system that collects information on mothers and their pregnancies of at least 20 weeks of gestational age, as well as babies with at least 400 g of birth weight. Maternal sociodemographic, health behavior, obstetric, labor, and delivery information are collected by attending midwives or medical practitioners.

For the purpose of this study, a sample of 2099 mothers who reported smoking tobacco during the first half of pregnancy, and who had live births between 2011–2019 in five public health facilities of Southern New South Wales Local Health District was utilized. Plural births (*n* = 31) were excluded from the analysis because compared with singletons, plural births may be biologically predisposed to low birth weight [24], and inclusion of plural births in the analysis could bias the results.

### 2.2. Study Setting

In 2011, under the Health Services Act 1997, the NSW government established Local Health Districts (LHDs) to deliver healthcare services across New South Wales [25]. NSW has eight metropolitans, and seven rural and regional LHDs. Southern NSW Local Health District (SNSWLHD) is one of the seven rural and regional LHDs with an estimated population of 205,281 [26,27]. Located in the southeastern part of NSW, SNSWLHD covers 44,534 square kilometers of land [28]. A wide range of maternity services [29] from antenatal, birthing, to postnatal care are provided from five public hospitals (Cooma Health Service in Cooma, Goulburn Base Hospital in Goulburn, Moruya District Hospital in Moruya, Queanbeyan Health Service in Queanbeyan, and South East Regional Hospital in Bega).

### 2.3. Study Outcome

In the public hospitals of SNSWLHD, the newborn’s first weight is usually measured within the first hour of life. The information on newborn’s weight in grams was used to construct a binary outcome coded as “1” if the newborn was measured <2500 g of birth weight or “0” otherwise.

### 2.4. Exposure Variables

To construct exposure variables, this study used four smoking behavior questions from the PDC, and some of these were previously described by Passmore and colleagues [30]. These questions were: (1) Did the mother smoke at all during the first half of pregnancy? (2) Did the mother smoke at all during the second half of pregnancy? (3) How many cigarettes each day on average were smoked in the first half of pregnancy? and (4) How many cigarettes each day on average were smoked in the second half of pregnancy?

During the first comprehensive antenatal care (ANC) visit in maternity units of SNSWLHD, the attending midwife collects various sociodemographic, obstetric, as well as the smoking-related information for pregnant women. In the section of perinatal data collection (PDC), under the sub-section of Body Mass Index, immunization, and smoking, in the electronic medical record (eMaternity), women are considered for smoking in the first half of pregnancy if they report smoking anytime within the first 20 weeks of gestational age. In addition, if identified as a smoker, a daily number of cigarettes are recorded as part of the smoking details. The first exposure variable of this study was smoking cessation during the second half of pregnancy. Smoking cessation during the second half of pregnancy was coded as ”1” if mothers reported smoking during the first half of pregnancy and continued smoking during the second half of pregnancy, and coded as “2” if mothers reported smoking during the first half of pregnancy but ceased smoking during the second half of pregnancy.

The second exposure variable of this study was coded as “1” if mothers ceased smoking during the second half of pregnancy, coded as “2” if mothers reported 1–10 average daily cigarettes during the second half of pregnancy, and coded as “3” if mothers reported >10 average daily cigarettes during the second half of pregnancy [31].

### 2.5. Confounding Variables

Except for a time-dependent confounder (year of birth), the remaining confounding variables included in this study were categorized into four distinct groups (sociodemographic, maternal or fetal, gestational, and health promotion). Sociodemographic variables were maternal socioeconomic status (SES) categorized as (high, middle, and low), maternal Aboriginal status (non-Aboriginal and Aboriginal), and hospital of birth (Cooma Health Service, Goulburn Base Hospital, Moruya District Hospital, Queanbeyan Health Service, and South East Regional Hospital). The Australian Bureau of Statistics has produced the Socio-Economic Index for Areas (SEIFA) indexes at the Statistical Area level by using a principal component analysis [32]. The SES variable in this study was constructed using the SEIFA Index of Relative Socio-Economic Advantage and Disadvantage. For the purpose of this study, the first 40% deciles of SES were arbitrarily referred to as high, the second 40% as middle, and the bottom 20% as low.

Maternal or fetal variables included in this study were mothers age at birth categorized as (<20 years, 20–34 years, and 35–49 years); history of previous pregnancy (2 or more previous pregnancies, 1 previous pregnancy, and no previous pregnancy); maternal chronic hypertension (yes and no); sex of baby (male and female); and type of delivery (normal vaginal, Caesarean section, and assisted vaginal). For the purpose of this study, forceps, vacuum extraction, and vaginal breech were considered for assisted vaginal.

Variables related to gestation were gestational diabetes (yes and no), and gestational hypertension (yes and no); health promotion variables were the number of antenatal care visits (<7 ANC visits, and ≥7 ANC visits) and duration of pregnancy at first ANC visit (<20 weeks and 20+ weeks). The categorization of number of antenatal care visits in this study is based on the minimum visits for ANC as recommended in NSW pregnancy care guidelines [33].

### 2.6. Statistical Analyses

The characteristics of the study sample was first described by calculating frequencies for each of the exposures and confounding variables and corresponding LBW with their 95% confidence intervals (CIs). Trends in prevalence of LBW and smoking cessation during the second half of pregnancy and their 95% confidence intervals (CIs) were then forest plotted.

The univariate logistic regression analysis was carried out to examine the independent association between exposure and outcome variables. To account for the complex hierarchical interrelationships between each of the different blocks of health determinants, this study used a five-stage hierarchical technique (Figure 1) similar to those described by Agho et al. [34]. As part of hierarchical technique, we first analyzed variables from the sociodemographic block (maternal SES category, maternal Aboriginal status, and hospital of birth) along with the year of birth to establish a baseline multivariate model (model 1). Maternal/fetal variables (maternal age at birth, history of previous pregnancy, chronic hypertension, sex of baby, and type of delivery) were then fitted into model 1 (model 2). Variables in model 2 were analyzed with gestational variables including gestational diabetes and gestational hypertension (model 3). In the next stage, health promotion variables (number of ANC visits, and duration of pregnancy at first ANC visit) were studied with model 3 (model 4). In the final model (model 5), the exposure variable within behavioral blocks (smoking cessation during the second half of pregnancy) was analyzed with model 4. All analyses were conducted in Stata version 16.1 (StataCorp, College Station, TX, USA).

### 2.7. Ethical Consideration

The Greater Western Human Research Ethics Committee in New South Wales, Australia approved this study (Protocol: Protocol_29.10.2020 V5.docx; Ethics Approval Number: 2020/ETH02384).

## 3. Results

### 3.1. Trends in Prevalence of Low Birth Weight

The prevalence of babies with LBW among mothers who reported smoking during pregnancy for the period (2011–2019) in SNSWLHD is almost eight percent (Figure 2). Even though the prevalence of LBW fluctuated between 2011 and 2019, the observed differences in fluctuation were not statistically significant.

### 3.2. Trends in Prevalence of Smoking Cessation

Prevalence of self-reported smoking cessation during the second half of pregnancy for the period 2011–2019 was estimated to be 17.40% (Figure 3). Prevalence of smoking cessation increased significantly (by 75%) between 2011 and 2014. Since 2014, fluctuation in smoking cessation occurred, with a 27% non-significant decrease in 2019 compared to 2014.

### 3.3. Characteristics of Study Population

Almost 17% of the study population reported an average consumption of >10 cigarettes a day (Table 1). Nearly 32% of mothers were from the low SES category. One in three mothers (31.5%) reported no previous pregnancy. Seventeen percent of mothers who reported smoking during pregnancy received <7 antenatal care visits. The percentage of the study sample from Cooma health service was lower (10.3%) compared to that of South East Regional Hospital (17.6%), Queanbeyan Health Service (19.2%), Goulburn Base Hospital (25.0%), and Moruya District Hospital (27.9%).

### 3.4. Impact of Smoking Cessation on Low Birth Weight

Multivariate logistic regression analyses showed that women who reported that they ceased smoking during the second half of pregnancy were 44% less likely to have a low birth weight baby (adjusted odds ratio (aOR) = 0.56; 95% CI: 0.34, 0.94, *p* = 0.028) compared to women who smoked throughout pregnancy (Table 2).

### 3.5. Dose-Response Analysis

A dose-response analysis revealed that LBW is significantly associated with mothers with an average daily smoking of 1–10 (aOR = 1.74; 95% CI: 1.03, 2.94, *p* = 0.037) or >10 cigarettes (aOR = 2.05; 95% CI: 1.12, 3.75, *p* = 0.020) during the second half of pregnancy compared to those who ceased smoking during the second half of pregnancy (Figure 4).

### 3.6. Protective and Risk Factors for Low Birth Weight

Belonging to the middle (aOR = 2.94; 95% CI: 1.00, 8.02, *p* = 0.050) or low (aOR = 4.04; 95% CI: 1.31, 12.44, *p* = 0.015) SES categories and advanced maternal age of 35–49 years (aOR = 1.87; 95% CI: 1.14, 3.05, *p* = 0.013) were risk factors for LBW; whereas, the attendance of seven or more ANC visits (aOR = 0.34; 95% CI: 0.23, 0.50, *p* =< 0.001) was protective against low birth weight (Table 2).

## 4. Discussion

Low birth weight babies among mothers who reported smoking during pregnancy in SNSWLHD remained constantly high over the study period (2011–2019). This study confirmed that smoking cessation during the second half of pregnancy prevents LBW in SNSWLHD. Despite the important health benefits, the smoking-cessation rate among pregnant women for the period 2011–2019 remained very low. A dose response analysis revealed that mothers with an average daily smoking of 1–10 cigarettes or >10 cigarettes during the second half of pregnancy were at risk of having babies of LBW compared to those who ceased smoking during the second half of pregnancy.

The protective effect of smoking cessation during pregnancy against LBW in this study is similar to those found in the birth certificate data from California’s Department of Public Health [17]. The analysis from the National Health Interview Survey [16] also found that mothers who ceased smoking in the first trimester of pregnancy were less likely to have babies of LBW compared to those who smoked beyond the first trimester of pregnancy (7.9% vs. 9.6%). Butler et al. urged that smoking cessation after the fourth month of pregnancy is critical to reduce LBW [35]. Nicotine, carbon monoxide, cadmium, and polycyclic aromatic hydrocarbons are forms of toxic components of cigarettes, and the reduced birthweight in our study among mothers who continued smoking throughout pregnancy may have resulted from prolonged antenatal exposure to harmful toxic components from cigarettes causing poor placental development and fetal growth [36,37,38]. The negative impact of smoking during pregnancy is not only limited to adverse perinatal outcomes such as LBW. The impact of smoking during pregnancy on postnatal health outcomes has been previously investigated and is well recognized [39,40]. The literature also suggests that the adverse postnatal health outcomes associated with maternal smoking during pregnancy may be mediated by smoking-induced LBW [41].

A dose-dependent relationship between tobacco smoking and LBW in this study is in agreement with previous research conducted in Australia [13], Taiwan [14], and Switzerland [15], which confirmed that a higher dose of cigarettes put pregnant women at greater risk of delivering LBW babies. However, the observed risk of LBW between two groups in this study (1–10 cigarettes a day and >10 cigarettes a day) did not differ statistically. Therefore, to prevent LBW, pregnancy care in health care services should emphasize a complete smoking cessation as opposed to cutting down on smoking, the latter has been previously documented as a common health belief among pregnant Australian women [42,43].

Our study found that the probability of babies with LBW among mothers in the older age bracket (35–49 years) was higher compared to that of women in the younger age group (20–34 years). The evidence of increased LBW babies among mothers in the older age bracket in this study is similar to those found previously [44,45]. It has been suggested that women with increasing maternal age are predisposed to a higher rate of pregnancy complications [46]; LBW among older mothers (34–49 years) in this study may be partly due to these complications [47]. Timely identification and management of pregnancy complications is a key aspect of quality pregnancy care. The provision of recommended antenatal care, therefore, is considered as an important pathway for providing the necessary care. In Australian public hospitals, the provision of antenatal care aims to promote a healthy pregnancy. Clinical practice guidelines for pregnancy care in Australia recommends a schedule of ten antenatal care visits for the first pregnancy without complications, and seven antenatal care visits for subsequent uncomplicated pregnancies [33]. The protective effect of at least seven ANC visits against LBW in this study is similar to the findings reported in the hospital-based Finnish study [48]. Even though the guideline for number of ANC visits differs between developing and developed countries, there is a consensus on findings that a lower or inadequate number of ANC visits is a risk factor for LBW [44,48]. The uptake of the recommended number of antenatal care visits is not only beneficial for the identification and management of pregnancy complications, but is regarded as a window of opportunity to reinforce behavioral-change communication to promote healthy behavior, including smoking cessation, and appropriate handling of sensitive issues such as domestic violence, which are keys to promoting a healthy pregnancy.

This study has policy and practice implications. Despite clear national guidelines, 17% pregnant women identified as smokers in this study did not attend at least 7 ANC visits. The non-use of the recommended number of antenatal care visits may be multifactorial, and residing in the regional geographic location is among those known for poor uptake of ANC [49]. Hence, promotion of smoking-cessation intervention through outreach services may help reduce LBW, especially for pregnant women where distance becomes a barrier for ANC visits. Given the trend of a lower prevalence of smoking cessation over the past several years, there is a need for initiatives aimed to improve systems to effectively drive the NSW smoking-cessation framework in maternity services. In addition, it is imperative to identify underlying causes to develop effective smoking-cessation strategies to suit the geographically diverse populations of SNSWLHD; to explore such causes, an in-depth anthropological study among women who continued smoking throughout pregnancy is required. Advanced maternal age is a risk factor for LBW in SNSWLHD. Hence, translation of this important knowledge to pregnant women during ANC visits can be an opportunistic healthcare strategy to promote age-specific future pregnancy that is protective against LBW.

This study has several strengths. First, the selected sample for this study was from a cohort of all births reported in SNSWLHD, which minimized the possibility of selection bias, and the findings from this study can be generalized to inform programmatic response to the wider district level population. Second, the data used in this study were pooled for the period 2011–2019 to increase sample size and statistical power. Third, the analysis accounts for a time-dependent confounder (year of birth), and thereby changes in maternal and child health care policies or programs over the study period (2011–2019) have been adjusted for in the risk model, thus, the findings from this study can be used to inform future district-level health practices, policies, and programs. However, this study is limited by the fact that important confounding variables for LBW such as maternal obesity and overweight [50], maternal anemia [51], maternal caffeine intake during pregnancy [52], and intimate partner violence [53,54] could not be adjusted for in the risk models because of the lack of data. The information on number of cigarettes collected in this study is based on mothers’ self-report of past events, which may have increased the recall or reporting bias. Another limitation is that because of the lack of data or small sample size, this study was not able to examine the association between LBW and smoking cessation that may have occurred in the early stage of pregnancy, or to compare LBW between mothers who did not smoke during the first half but smoked during the second half of pregnancy (non-smoker to smoker) with mothers who smoked during the first but not during the second half of pregnancy (smoker to non-smoker). Therefore, future comparative research on smoking in different stages of pregnancy and birthweight would be helpful to understand the association further to direct the smoking-cessation effort. In addition, the data source of this study (Perinatal Data Collection) does not have information on postnatal child health outcomes. Future studies on postnatal health outcomes that may be mediated by smoking induced LBW can shed more light on the topic to help understand the magnitude of postnatal adverse health outcomes attributed to smoking during pregnancy.

## 5. Conclusions

This study concludes that smoking cessation during the second half of pregnancy in Southern New South Wales remained very low for the period 2011–2019. To prevent LBW, an effective smoking-cessation intervention is urgently required for pregnant women who smoke. As part of protective strategy, health promotion initiatives should promote antenatal care visits of at least seven or more and target tailored intervention for risk population such as mothers with low SES and those in an older age bracket (35–49 years).

## Figures and Tables

**Figure 1 ijerph-18-03417-f001:**
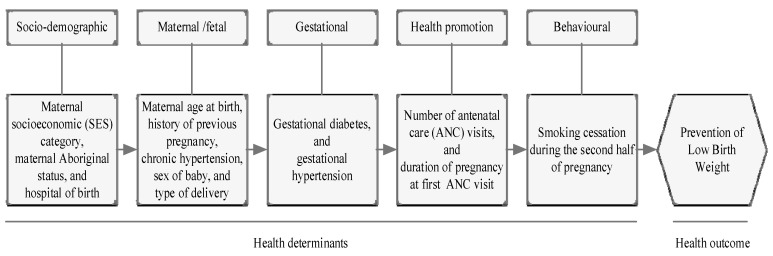
Conceptual framework for the analysis of smoking cessation during the second half of pregnancy and low birth weight (LBW) of babies among mothers of reproductive age (15–49 years).

**Figure 2 ijerph-18-03417-f002:**
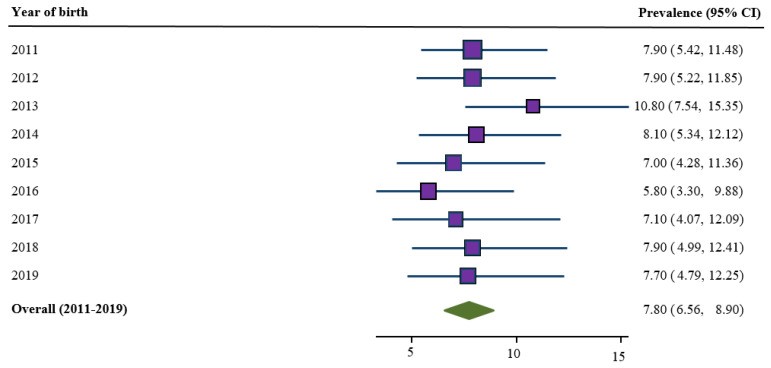
Trends in prevalence of LBW for the period 2011–2019, Southern New South Wales Local Health District (SNSWLHD) (*N* = 2099).

**Figure 3 ijerph-18-03417-f003:**
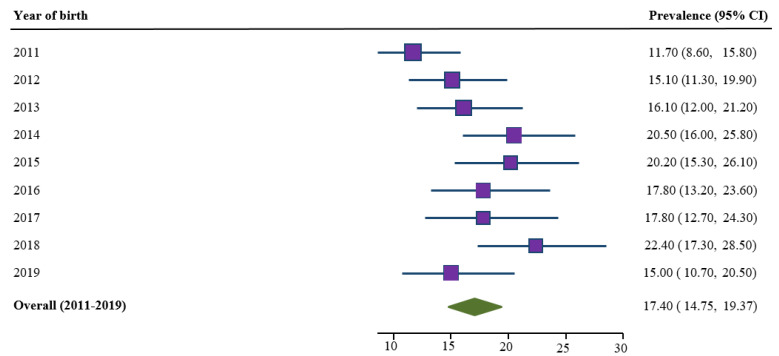
Trends in prevalence of smoking cessation during the second half of pregnancy for the period 2011–2019, SNSWLHD (*N* = 2099).

**Figure 4 ijerph-18-03417-f004:**
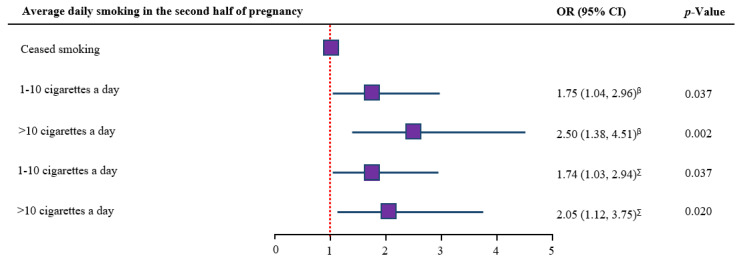
Smoking dose and low birth weight among pregnant smokers in SNSWLHD (2011–2019). ^β^ unadjusted odds ratio; ^∑^ adjusted odds ratio. Model adjusted for year of birth, socio-demographic (maternal SES category, maternal Aboriginal status, hospital of birth), maternal/fetal (maternal age at birth, history of previous pregnancy, chronic hypertension, sex of baby, type of delivery), gestational (gestational diabetes, gestational hypertension), and health promotion (number of ANC visits, duration of pregnancy at first ANC visit) variables; CI = confidence interval.

**Table 1 ijerph-18-03417-t001:** Characteristics of the study population, number of babies with low birth weight, and percentage with 95% confidence interval (CI) in SNSWLHD (*N* = 2099).

Characteristics		Low Birth Weight	
Confounding variables	N (%)	n	% (95% CI)
Maternal SES category (*N* = 2096)			
High	208 (9.9)	5	2.4 (1.0, 5.6)
Middle	1220 (58.1)	98	8.0 (6.6, 9.7)
Low	668 (31.8)	63	9.4 (7.4, 11.9)
Maternal Aboriginal status (*N* = 2096)			
Non-aboriginal	1745 (83.1)	129	7.4 (6.3, 8.7)
Aboriginal	351 (16.7)	37	10.5 (7.7, 14.2)
Hospital of birth			
Cooma Health Service	217 (10.3)	14	6.5 (3.9, 10.6)
Goulburn Base Hospital	524 (25.0)	42	8.0 (6.0, 10.7)
Moruya District Hospital	586 (27.9)	58	9.9 (7.7, 12.6)
Queanbeyan Health Service	403 (19.2)	19	4.7 (3.0, 7.3)
South East Regional Hospital	369 (17.6)	33	8.9 (6.4, 12.3)
Maternal age at birth			
<20 years	207 (9.9)	15	7.2 (4.4, 11.7)
20–34 years	1647 (78.5)	122	7.4 (6.2, 8.8)
35–49 years	245 (11.7)	29	11.8 (8.3, 16.5)
History of previous pregnancy (*N* = 2093)			
2 or more previous pregnancies	856 (40.8)	71	8.3 (6.6, 10.3)
1 previous pregnancy	576 (27.4)	41	7.1 (5.3, 9.5)
no previous pregnancy	661 (31.5)	54	8.2 (6.3, 10.5)
Sex of baby			
Male	1069 (50.9)	78	7.3 (5.9, 9.0)
Female	1030 (49.1)	88	8.5 (6.9, 10.3)
Type of delivery			
Normal vaginal	1436 (68.4)	111	7.7 (6.5, 9.2)
Caesarean section	490 (23.3)	42	8.6 (6.4, 11.4)
Assisted vaginal	173 (8.3)	13	7.5 (4.4, 12.5)
Diagnosed with gestational diabetes (*N* = 2094)			
No	1990 (94.8)	158	7.9 (6.8, 9.2)
Yes	104 (4.9)	8	7.7 (3.9, 14.6)
Diagnosed with gestational hypertension			
No	2060 (98.1)	161	7.8 (6.7, 9.1)
Yes	39 (1.9)	5	12.8 (5.4, 27.3)
Number of antenatal care visits (*N* = 2071)			
<7 visits	357 (17.0)	53	14.8 (11.5, 18.9)
7+ visits	1703 (81.1)	102	6.0 (4.9, 7.2)
Duration of pregnancy at first antenatal visit (*N* = 2082)			
<20 weeks	1365 (65.0)	106	7.8 (6.5, 9.3)
20+ weeks	717 (34.2)	53	7.4 (5.7, 9.6)
Exposure variables			
Smoking cessation during the second half of pregnancy			
Continued smoking	1740 (82.9)	149	8.6 (7.3, 10.0)
Ceased smoking	359 (17.1)	17	4.7 (3.0, 7.5)
Average daily smoking in the second half of pregnancy (*N* = 2073)			
Ceased smoking	359 (17.1)	17	4.7 (3.0, 7.5)
1–10 cigarettes a day	1361 (64.8)	109	8.0 (6.7, 9.6)
>10 cigarettes a day	353 (16.8)	39	11.0 (8.2, 14.7)

**Table 2 ijerph-18-03417-t002:** Unadjusted and adjusted odds ratio (aOR) and 95% confidence interval (CI) for study variables and low birth weight (LBW) in SNSWLHD (2011–2019).

Characteristics	Unadjusted Model		Model 1		Model 2		Model 3		Model 4		Model 5	
Confounding Variables	OR (95% CI)	*p*-Value	aOR (95% CI)	*p*-Value	aOR (95% CI)	*p*-Value	aOR (95% CI)	*p*-Value	aOR (95% CI)	*p*-Value	aOR (95% CI)	*p*-Value
Year of birth												
2011	1.00		1.00		1.00		1.00		1.00		1.00	
2012	1.18 (0.61, 2.26)	0.619	1.21 (0.62, 2.34)	0.578	1.23 (0.63, 2.40)	0.537	1.23 (0.63, 2.40)	0.539	1.31 (0.66, 2.59)	0.433	1.35 (0.68, 2.66)	0.389
2013	1.74 (0.94, 3.21)	0.078	1.70 (0.91, 3.17)	0.096	1.76 (0.93, 3.31)	0.081	1.74 (0.93, 3.28)	0.085	1.97 (1.03, 3.74)	0.039	2.02 (1.06, 3.84)	0.033
2014	1.20 (0.62, 2.30)	0.584	1.16 (0.60, 2.23)	0.665	1.16 (0.60, 2.25)	0.654	1.16 (0.60, 2.24)	0.652	1.31 (0.67, 2.57)	0.425	1.40 (0.72, 2.75)	0.323
2015	1.09 (0.54, 2.20)	0.812	1.05 (0.51, 2.16)	0.887	1.06 (0.52, 2.18)	0.873	1.06 (0.51, 2.17)	0.882	1.27 (0.61, 2.64)	0.516	1.34 (0.64, 2.78)	0.437
2016	0.80 (0.37, 1.73)	0.574	0.75 (0.35, 1.62)	0.466	0.78 (0.36, 1.68)	0.52	0.78 (0.36, 1.69)	0.53	0.9 (0.42, 1.95)	0.789	0.93 (0.43, 2.01)	0.852
2017	1.10 (0.52, 2.34)	0.797	1.08 (0.51, 2.31)	0.844	1.05 (0.49, 2.24)	0.895	1.05 (0.49, 2.24)	0.904	1.09 (0.51, 2.33)	0.827	1.12 (0.52, 2.40)	0.779
2018	1.18 (0.59, 2.35)	0.639	1.14 (0.57, 2.30)	0.713	1.11 (0.55, 2.26)	0.766	1.12 (0.55, 2.28)	0.749	1.24 (0.6, 2.55)	0.566	1.31 (0.63, 2.70)	0.469
2019	1.13 (0.56, 2.28)	0.734	1.08 (0.53, 2.20)	0.821	1.05 (0.52, 2.14)	0.885	1.06 (0.52, 2.18)	0.867	1.19 (0.58, 2.45)	0.627	1.22 (0.59, 2.50)	0.595
Maternal SES category												
High	1.00		1.00		1.00		1.00		1.00		1.00	
Middle	4.12 (1.50, 11.36)	0.006	2.96 (0.99, 8.82)	0.051	2.92 (0.98, 8.66)	0.054	2.92 (0.98, 8.66)	0.054	2.97 (1.01, 8.78)	0.049	2.94 (1.00, 8.62)	0.050
Low	5.05 (1.81, 14.09)	0.002	3.92 (1.25, 12.26)	0.019	4.01 (1.29, 12.5)	0.017	3.99 (1.28, 12.42)	0.017	4.15 (1.34, 12.87)	0.014	4.04 (1.31, 12.44)	0.015
Maternal Aboriginal status												
Non-Aboriginal	1.00		1.00		1.00		1.00		1.00		1.00	
Aboriginal	1.48 (0.99, 2.20)	0.057	1.38 (0.92, 2.07)	0.117	1.50 (0.99, 2.27)	0.058	1.49 (0.98, 2.26)	0.062	1.39 (0.91, 2.13)	0.131	1.36 (0.89, 2.10)	0.156
Hospital of birth												
Cooma Health Service	1.00											
Goulburn Base Hospital	1.18 (0.63, 2.23)	0.605	0.84 (0.41, 1.71)	0.622	0.84 (0.41, 1.72)	0.629	0.83 (0.41, 1.72)	0.622	0.73 (0.35, 1.51)	0.394	0.75 (0.37, 1.55)	0.443
Moruya District Hospital	1.50 (0.82, 2.76)	0.192	1.13 (0.60, 2.13)	0.707	1.12 (0.59, 2.14)	0.724	1.12 (0.59, 2.13)	0.737	0.96 (0.5, 1.86)	0.912	0.99 (0.52, 1.91)	0.986
Queanbeyan Health Service	0.57 (0.27, 1.22)	0.147	0.65 (0.30, 1.43)	0.286	0.66 (0.30, 1.45)	0.299	0.66 (0.30, 1.44)	0.296	0.61 (0.27, 1.36)	0.226	0.61 (0.27, 1.35)	0.220
South East Regional Hospital	1.39 (0.72, 2.66)	0.326	1.12 (0.58, 2.15)	0.744	1.09 (0.56, 2.11)	0.799	1.08 (0.56, 2.09)	0.814	0.95 (0.48, 1.85)	0.873	0.96 (0.49, 1.87)	0.902
Maternal age at birth												
20–34 years	1.00				1.00		1.00		1.00		1.00	
<20 years	0.96 (0.54, 1.70)	0.88			0.78 (0.42, 1.47)	0.446	0.79 (0.42, 1.48)	0.462	0.72 (0.38, 1.37)	0.322	0.69 (0.36, 1.30)	0.250
35–49 years	1.69 (1.08, 2.63)	0.021			1.73 (1.08, 2.78)	0.023	1.73 (1.07, 2.80)	0.027	1.88 (1.15, 3.07)	0.012	1.87 (1.14, 3.05)	0.013
History of previous pregnancy												
2 or more previous pregnancy	1.00				1.00		1.00		1.00		1.00	
1 previous pregnancy	0.81 (0.53, 1.22)	0.308			0.98 (0.63, 1.52)	0.92	0.97 (0.63, 1.50)	0.898	1 (0.64, 1.58)	0.983	1.01 (0.64, 1.58)	0.975
no previous pregnancy	0.91 (0.62, 1.34)	0.645			1.22 (0.77, 1.94)	0.396	1.20 (0.76, 1.91)	0.437	1.39 (0.86, 2.27)	0.179	1.46 (0.90, 2.36)	0.126
Diagnosed with chronic hypertension												
No	1.00				1.00		1.00		1.00		1.00	
Yes	2.73 (0.58, 12.76)	0.202			2.21 (0.44, 11.18)	0.338	2.26 (0.45, 11.49)	0.324	2.98 (0.56, 15.9)	0.2	3.32 (0.57, 19.41)	0.182
Sex of baby												
Male	1.00				1.00		1.00		1.00		1.00	
Female	1.22 (0.88, 1.69)	0.239			1.23 (0.88, 1.71)	0.224	1.22 (0.88, 1.71)	0.23	1.27 (0.91, 1.77)	0.155	1.27 (0.91, 1.77)	0.161
Type of delivery												
Normal vaginal	1.00				1.00		1.00		1.00		1.00	
Caesarean section	1.17 (0.80, 1.71)	0.414			1.14 (0.76, 1.69)	0.533	1.13 (0.76, 1.69)	0.538	1.13 (0.75, 1.7)	0.556	1.15 (0.76, 1.74)	0.497
Assisted vaginal	0.87 (0.46, 1.66)	0.678			0.85 (0.43, 1.67)	0.628	0.85 (0.43, 1.68)	0.642	0.82 (0.41, 1.62)	0.563	0.87 (0.44, 1.73)	0.688
Diagnosed with gestational diabetes												
No	1.00						1.00		1.00		1.00	
Yes	1.02 (0.48, 2.13)	0.967					0.89 (0.40, 1.96)	0.764	1 (0.45, 2.2)	0.996	0.99 (0.45, 2.16)	0.975
Diagnosed with gestational hypertension												
No	1.00						1.00		1.00		1.00	
Yes	1.82 (0.70, 4.72)	0.22					1.40 (0.54, 3.67)	0.492	1.48 (0.54, 4.08)	0.445	1.49 (0.53, 4.16)	0.445
Number of antenatal care visits												
<7 visits	1.00								1.00		1.00	
7+ visits	0.37 (0.26, 0.52)	<0.001							0.33 (0.22, 0.49)	<0.001	0.34 (0.23, 0.50)	<0.001
Duration of pregnancy at first antenatal visit												
<20 weeks	1.00								1.00		1.00	
20 + weeks	1.06 (0.75, 1.50)	0.743							1.26 (0.86, 1.86)	0.235	1.28 (0.86, 1.88)	0.221
Exposure variables												
Smoking cessation during the second half of pregnancy												
Continued smoking	1.00										1.00	
Ceased smoking	0.57 (0.34, 0.95)	0.031									0.56 (0.34, 0.94)	0.028

## Data Availability

Access to the data used in this study was in accordance with the research protocol submitted to the Greater Western Human Research Ethics Committee. For data enquiries, please contact Southern New South Wales Local Health District (SNSWLHD) Research Governance Officer, Postal Address: 34 Lowe Street, Queanbeyan, New South Wales, Australia 2620, Tel: +61-02-6150-7574; Mob +61-477-329-161, Email: Dot.hughes@health.nsw.gov.au. Greater Western Human Research Ethics Committee (GWHREC) Research Governance Officer, Postal Address: PO Box 143, 39 Hampden Park Road, Bathurst, Australia 2795 Telephone: +61-02-6330-5948, Email: Phil.Sanders@health.nsw.gov.au.

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
