# Peer review of "Smoking Cessation during the Second Half of Pregnancy Prevents Low Birth Weight among Australian Born Babies in Regional New South Wales"

_ijerph, 2021, doi:10.3390/ijerph18073417_

Round 1
Reviewer 1 Report
In the current manuscript the association between smoking cessation in the second half of pregnancy and birth weight is assessed. This is a very well-written and well-structured manuscript and the results are potentially very important as they underline the need to direct cessation efforts at all stages of pregnancy.
However, there is one major point that I am still unclear about: I could not find any information on how exactly women were categorized as smoking in the first half of pregnancy. It would be important to know whether these were women who smoked during most of the first half or not. Importantly if most of these women were smoking in very early pregnancy only (potentially before they knew about the pregnancy) and then stopped a much lower effect on birthweight would be expected. Information about how smoking during the first half was defined (did they have had to smoke one cigarette to count as smokers?) and information on daily smoking in the first half of pregnancy and how smoking during first half of pregnancy compared between those that quit in the second half and those that did not would be extremely important in order to understand the magnitude of these findings.
Reviewer 2 Report
It is a well written paper. Only few minor comment for the manuscript.
1, Why using the daily smoking of 10 cigarettes as cutoff in the analysis.
2, Most of the studies use the trimester of pregency in the analysis, why this study use the second half of pregency?
3, It is better to add evidence in some developing countries in the discussion.
Round 2
Reviewer 1 Report
All comments have been addressed! Thanks a lot!
